# The Thematic Analysis of Barriers to Immediate Post-Partum Long-Acting Reversible Contraception

**DOI:** 10.3390/healthcare12222208

**Published:** 2024-11-05

**Authors:** Mahwish Iqbal, Tayyiba Wasim, Saeed A. AlQahtani, Anwar A. Alghamdi, Aftab Ahmad, Ahmad Hefnawy Abbas, Natasha Bushra, Usman Thattarauthodiyil, Vigneshwaran Easwaran, Muhammad Afzal, Narayana Goruntla, Nehmat Ghaboura, Mohammad Jaffar Sadiq Mantargi

**Affiliations:** 1Naseem Jeddah Medical Center, Jeddah 23342, Saudi Arabia; drmahwishiqbal46@gmail.com; 2Obstetrics and Gynaecology, Services Institute of Medical Sciences, Lahore 54000, Pakistan; tayyibawasim@yahoo.com; 3Department of Clinical Practice, College of Pharmacy, Jazan University, Jazan 45142, Saudi Arabia; saqahtani@jazanu.edu.sa; 4Health Information Technology Department, The Applied College, King Abdulaziz University, Jeddah 21589, Saudi Arabia; nloalgamdi7@kau.edu.sa (A.A.A.); abdulsalam@kau.edu.sa (A.A.); 5Pharmacovigilance and Medication Safety Unit, Center of Research Excellence for Drug Research and Pharmaceutical Industries, King Abdulaziz University, Jeddah 21589, Saudi Arabia; 6Clinical Toxicology, Department of Forensic Medicine and Clinical Toxicology, Faculty of Medicine, Minia University, Minya 61519, Egypt; dr_hefnawyatox77@minia.edu.eg; 7Obstetrics and Gynaecology, Post Graduate Institute of Medical Sciences, Lahore 54000, Pakistan; natashausman5@hotmail.com; 8Physiotherapy Program, Department of Physiotherapy, Batterjee Medical College, P.O. Box 23819, Jeddah 21442, Saudi Arabia; usmuptb@gmail.com; 9Department of Clinical Pharmacy, College of Pharmacy, King Khalid University, Abha 61421, Saudi Arabia; vbagyalakshmi@kku.edu.sa; 10Department of Pharmaceutical Sciences, Pharmacy Program, Batterjee Medical College, P.O. Box 23819, Jeddah 21442, Saudi Arabia; mohmmad.afzal@bmc.edu.sa; 11Department of Clinical Pharmacy and Pharmacy Practice, School of Pharmacy, Kampala International University, Western Campus, Ishaka 20000, Uganda; narayanagoruntla@gmail.com; 12Pharmacy Program, Department of Pharmacy Practice, Batterjee Medical College, P.O. Box 23819, Jeddah 21442, Saudi Arabia; pharmacy8.jed@bmc.edu.sa

**Keywords:** contraceptives, LARC, family planning, implants, intrauterine devices, family planning

## Abstract

Background: Globally, many women express the desire to avoid immediate pregnancy for 24 months postdelivery, and only forty percent use contraceptives during this period. There is an enormous demand for postpartum family planning, particularly in developing countries with low- or middle-income grades. Postpartum intrauterine devices such as long-acting reversible contraceptives (LARCs) are among the most effective methods of family planning in the immediate postpartum period, yet their effectiveness is hindered because of a lack of availability and training. Strategies to increase access to LARCs are essential. Hence, the purpose of the current study is to determine the barriers among healthcare providers in providing immediate postpartum family planning services. Methods: A cross-sectional study was conducted in the Department of Obstetrics and Gynaecology at SIMS, a tertiary care teaching hospital, from January to March 2024. Approximately 293 healthcare providers who fulfilled the inclusion criteria were provided an online questionnaire in the form of a Google Forms link, which included a structured questionnaire focusing on various aspects, including demographics, knowledge, practices, and barriers in their practice. The data collected were analysed through SPSS version 26, which employs chi-square tests and Pearson’s correlation to determine any significant associations among them. Based on the key statistical outcomes and the significant correlations observed related to the data, a thematic analysis through SWOT (strengths, weaknesses, opportunities, threats) was conducted. The study adhered to the method outlined by Braun and Clarke (2006) and compiled with the COREQ (consolidated criteria for reporting qualitative research) checklist to uphold methodological integrity. Results: Among the participants, 92.4% provided family planning counselling after childbirth, predominantly during the antenatal period (75.1%), and the provision of immediate postnatal family planning was reported in 76.1% of the participants, with PPIUDs identified as the most utilized method by 52.6%. Various barriers were identified, including insufficient training on Implanon (33.4%) and the PPIUCD (12.6%), the unavailability of implants (59.0%), and a lack of interest among patients (46.1%). Statistically significant associations were observed between the practice setting and knowledge of postpartum family planning (*p* = 0.002), as well as deficiencies in training for the PPIUCD (*p* < 0.001). The study highlights the place of practice and the practitioners’ experience as significant strengths in offering immediate postpartum contraception and referring patients for family planning. However, qualification was identified as a limiting factor for practicing immediate postpartum family planning. Conclusions: This study revealed significant difficulty in delivering prompt postpartum long-acting reversible contraceptives (LARCs), underscoring the necessity of improved education and training for professionals. Focusing on these challenges is important in enhancing postpartum family planning acceptance and decreasing unfulfilled requirements in resource-limited settings.

## 1. Introduction

Globally, 95% of women wish to avoid pregnancy in the first 24 months after childbirth; however, only approximately 40% will have used contraception during this period [1,2]. The unmet need for postpartum family planning (PPFP) is much greater in low- and middle-income countries. In Pakistan, the unmet need for PPFP is more than 50% among women in the immediate postpartum period [3]. Postpartum intrauterine devices (PPIUDs) are effective and affordable. The PPFP method can reduce unmet needs [4]. Health providers play a key role in addressing unmet needs by providing quality and timely PPIUD services [5,6]. However, maintaining clinical standards is crucial, and as such, quality training and mentoring are necessary [7,8]. Postpartum family planning is a compelling concern of global importance because of its salience to unplanned pregnancies and to maternal and infant health in developing countries. However, women face the highest level of unmet need for contraception in the year following birth [9]. A cost-effective way to inform women about their risk of becoming pregnant after the birth of a child is to integrate family planning counselling and services with maternal and infant health services [10].

Strategies to increase access to long-acting reversible contraceptives (LARCs) are essential. Prior studies identified knowledge gaps regarding patient eligibility for LARCs [11] as well as practice differences at the provider level. Known barriers to LARC provision include provider training and the inability to perform same-day insertion, which together limit overall use [12,13].

Unfortunately, only approximately one-third of women who desire postpartum LARCs will ultimately obtain it by 8–12 weeks postpartum if they do not obtain it before hospital discharge [14]. This has been demonstrated with regard to both intrauterine devices and contraceptive implants. These women are not only at risk for unintended pregnancy but also at risk for a short interpregnancy interval, even if they are given an alternative contraceptive in the interim [15].

Among five countries in Latin America and the Caribbean, approximately half of pregnant women receive some family planning information during antenatal care visits [16]. In sub-Saharan Africa, the proportion of postpartum women who are exposed to the risk of pregnancy by having sex while using no family planning method within two years after childbirth is nearly one-third [17].

Closely spaced pregnancies within the first-year postpartum increase the risk of death for both the mother and the baby. Many countries recommend providing pregnant women with postpartum family planning counselling during antenatal care visits [18]. However, data on the extent to which providers utilize these opportunities and the role of family planning counselling provided during antenatal care in promoting modern postpartum family planning remain limited, particularly in developing countries. This study aims to explore the training-based knowledge, religious and cultural barriers, the impact of education, the availability of IUCD, and how beneficial policies affect practitioners, which makes the study unique [19]. Therefore, this study aims to identify the barriers among healthcare providers in providing immediate postpartum family planning services.

## 2. Materials and Methods

The current cross-sectional research was performed at the Department of Obstetrics and Gynaecology, SIMS, Lahore, a tertiary care teaching hospital. The sample size [20] was determined on the basis of parameters including the 95% confidence interval and 5% margin of error, with the estimate that the target population consists of 48,000 active physicians in the region. Furthermore, assuming a 50% response rate of the consented participants, the required number of inclusions was 283; however, 293 participants’ responses were selected, as those were considered complete from an additional 20 included as buffers in the case of incomplete participation [21].

The inclusion criteria for the current investigation include practicing healthcare professionals in the fields of obstetrics and gynaecology providing services across different geographical settings, thus demonstrating an intention to include a rich cohort of required healthcare providers in the study. Simultaneously, healthcare providers who were not involved in offering services in obstetrics and gynaecology were excluded from the study. The exclusion criteria were crucial for maintaining the integrity of the study. This approach ensures the relevance and applicability of the research outcomes to specific research questions [22].

The methodology strategy facilitates a focused examination of obstacles to immediate postpartum long-acting reversible contraceptives (LARCs) among obstetric healthcare providers. By focusing on well-defined samples and the application of precise statistical tools, this study aims to generate dependable and broadly applicable insights into identifiable research issues [23]. Lahore/SIMS offers a favourable setting for data collection and ensures the availability of diverse groups of obstetric healthcare providers.

### 2.1. Ethical Approval

Batterjee Medical College’s institutional review board (IRB) has reviewed and approved the research proposal submitted prior to the beginning of the study with the suitable recommendations in accordance with BMC-IRB (research proposal number: RES-2023-0089; ethical review panel authentication code: HP-02-J-113).

### 2.2. Study Tool

The study questionnaire was prepared by referring to various sources available online that evaluate the knowledge, attitudes, and practice domains which fit the criteria of obstetrics and gynaecology. The initial questionnaire prepared underwent an expert review involving the authors of the study and the final version was drafted. It consists of four domains, i.e., consent and demographics, seven questions related to knowledge, three questions related to training, three questions related to side effects, five questions related to workload, three questions related to financial barriers, and two questions related to religious barriers. A trial survey was conducted to study the internal consistency by determining the Cronbach’s alpha, and adjustments were included on the basis of the results of the pilot test. The final version after the approval of all the authors was transferred to Google Forms, and the link generated was distributed to the participants for their answers via email, WhatsApp, institutional internal communications, telephonic requests, and sharing links via SMS [24].

### 2.3. Data Collection

The data were collected from the month of January 2024 to March 2024, with a regular check-up of the Google form for identifying the number of hits during the given time period. Only a single attempt per individual was made to avoid the occurrence of bias. At the end of March, the Excel spreadsheet bearing the data of the participants was downloaded from Google Forms for further statistical processing [25].

### 2.4. Statistical Analysis

The data obtained through the questionnaire were entered into the Microsoft Excel spreadsheet, coded accordingly and saved as a .csv file. This information was uploaded into IBM SPSS v.26, and the codes mentioned in the spreadsheet were labelled for ease of data processing. The questionnaire was divided into three sections: demographic details of the study participants, items representing the participants’ knowledge of postpartum family planning, and postpartum family planning methods. The nominal variables, such as age, number of deliveries per day, and number of deliveries per month, were converted into the ordinal format, and the descriptive statistics of the data were obtained and are expressed in Table 1, Table 2 and Table 3. Chi-square analysis of the data was performed to identify the significant associations of different variables, and Pearson’s correlation was chosen to justify the strengths, weaknesses, opportunities, and threats as a part of thematic analysis [21]. *p* * values ≤ 0.05 were considered significant, *p* ** values ≤ 0.01 were considered highly significant, and *p* *** values ≤ 0.001 were considered highly significant.

### 2.5. Thematic Analysis (SWOT Analysis)

The study followed a six-phase approach to thematic analysis as explained by Braun and Clarke (2006) [26]. The steps involved are data familiarisation, initial code generation, exploring themes, reviewing themes, defining themes, and compiling a report. The data was analysed manually to allow an in-depth interpretation of the results.

The questionnaire employed in the study consists of quantitative and qualitative questions. The quantitative (closed-ended questions) multiple-choice questions are designed to facilitate easier analysis, comparison, and statistical evaluation, which could be answered at ease and pace of the answerer. However, qualitative questions (open-ended) allow the respondents to express information which might be limited by closed-ended questions. This question may capture deeper insight into their experiences. The combination of different type of questions gives the benefit of both precision in measurement and flexibility in understanding responses. The qualitative questions included in the study are as follows: When do you offer family planning counselling to patients? What challenges do you face while considering contraceptives? Drawbacks of contraceptive usage postpartum. The questions were aimed to facilitate thematic analysis.

Systematic coding by a repetitive methodology, where initial codes were identified based on recurrent concepts in the responses was used. The coding was inductive, i.e., emerging from data, which were then organized into broader themes representing key aspects. The coding was performed by two authors with expertise in qualitative research, and to establish consistency and reliability, a comparative analysis along with discussion was performed. Any inconsistencies were resolved through discussion and timely, resulting in mutual understanding. Finally, a SWOT criterion was preferred [27].

The authors have reviewed the manuscript and confirm adherence to the COREQ guidelines through 32 item checklists. The details regarding the research team, reflexivity, study design, and data analysis are in accordance with COREQ criteria [28].

On the basis of the statistical results, the highly significant values related to the provider’s attitude and institutional aspects were considered strengths, the significant values with inadequate training and patient education were categorized as weaknesses, the significant values reflecting community outreach programs and supportive policy amendments were considered opportunities, and significant values representing cultural barriers, costs, and expenses were considered threats/drawbacks which were the results of simplifying the themes obtained [29].

## 3. Results

The study included a total of 293 participants ranging from 24–69 years of age, of which 67.9% belonged to the age band of 20–29 years, with the highest qualification of 54.6%. Among the different options, teaching hospitals were selected by 57.7% of the participants as their professional places of work, which are located in urban inner cities (51.2). As many as 98.0% of the participants expressed performing 0–50 deliveries per day and 58.4% deliveries per month, even though the number of deliveries ranged above 3000 deliveries per month from a few of the responses. The majority of the study participants are relatively recent graduates of their professional setups, with 47.8% having 1–5 years of professional experience [Table 1].

The nature and impact of postpartum family planning counselling was explained in Table 2, where a total of 94.2% of the study participants agreed on offering family planning counselling to patients, 75.1% agreed on providing the same counselling during the antenatal period, and 7.2% did not provide patient counselling, as the patients were not interested in the details of family planning. As high as 82.3% of practitioners failed to complete their answer as to why they do not offer family planning counselling or recommend usage of contraceptives to the patients immediately postpartum, which creates a large vacuum in the applicability of knowledge and skills developed throughout their educational years.

A total of 76.1% agreed with providing postpartum family planning through the IUCD, as it was the doctor’s choice (52.6%) as well as the patient’s choice (40.6%). A total of 33.4% agreed that they do not have proper training to Implanon, 12.6% agreed that they lack proper training to PPIUCD, 59.0% expressed that the unavailability of implants was the issue, and for 46.1%, it was due to a lack of patient interest. These are the major challenges faced. A total of 82.9% expressed that LARCs are not contraindicated in breast feeding, and 80.5% agreed that they are not contraindicated immediately post placental. Hence, the results confirmed that LARCs are the best option available in the case of postpartum family planning modalities [Table 3].

Age, qualifications, place of practice, duration of practice, number of deliveries per day, and number of deliveries per month were strongly and positively associated with the study questions, of which offering immediate postpartum family planning to patients was the highly significant outcome. This shows the commitment and dedication of the professionals included in the current study. A strong correlation between qualification, place of practice, and the provision of family planning counselling (X^2^ = 0.000) was identified, alongside significant obstacles such as inadequate training for Implanon handling (X^2^ = 0.038), training of insertion of IUCD (X^2^ = 0.00008), and PPIUCD handling (X^2^ = 0.000). Knowledge and familiarities with immediate postpartum family planning present inconclusive outcomes, further recommending the need for training and resources [Table 4].

The following description presents the values of 2-tailed Pearson’s correlation as factors analysed (r = Pearson’s correlation, *p* = significance).

The study’s strengths, weaknesses, opportunities, and threats were identified according to their significance and Pearson’s correlation coefficient, from which age, qualifications, place of work, and duration of work compared with the knowledge, attitudes, and practices of postpartum contraception were derived. Healthcare providers situated in urban areas are interested in implementing these services (r = 0.185 **, *p* = 0.002; r = 0.261 **, *p* = 0.000). Moreover, the duration of practice and advanced qualifications are positively associated with refereeing patients for family planning (r = 0.223 **, *p* = 0.004; r = 0.180 *, *p* = 0.022). Furthermore, knowledge of the application of LARCs in patients with a history of ectopic pregnancy or during breast feeding was high (r = 0.253, *p* = 0.000; r = 0.228, *p* = 0.000), which was identified as a strength of the study participants.

However, practitioners with higher qualifications did not consider counselling for family planning (r= −0.302 **, *p* = 0.000; r= −0.122 *, *p* = 0.044), and challenges related to time constraints, the unavailability of IUCDs, and inadequate training in insertion techniques further hinder the practice of family planning (r = 0.223 **, *p* = 0.010; r = -−0.257 **, *p* = 0.002; r = 0.199 *, *p* = 0.014).

Training deficiencies and a continuous supply of contraceptives are considered opportunities (r = −0.129 *, *p* = 0.034; r = −0.257 **, *p* = 0.000; r = −0.121*, *p* = 0.044).

Potential challenges include the lack of knowledge of immediate postpartum family planning among highly skilled practitioners (r = −0.159 **, *p* = 0.007) and the substantial training gap regarding postpartum intrauterine contraceptive devices (PPIUCDs) (r = 0.434 **, *p* = 0.000; r = 0.275 **, *p* = 0.001) are expressed as threats/drawbacks [Table 5].

## 4. Discussion

The current study highlights the attitudes of medical practitioners toward the practice of postpartum family planning through a questionnaire-based survey and presents a thematic analysis of the results for a better understanding of the strengths, weaknesses, opportunities, and threats (SWOT) related to postpartum family planning from the perspective of expert clinicians and related decision-making. Various attempts have been made to study the attitudes of clinicians related to postpartum family planning, but these attempts consist of diverse geographical areas [30,31]. Hence, in parallel, an attempt has been made to study the barriers and derive a SWOT analysis of the barriers related to postpartum family planning clinically following COREQ checklist and deriving a SWOT-based thematic analysis following the idea explained by Braun and Clarke (2006).

Younger healthcare providers adopt novel research methods and align with the usage of newer techniques in medical practice, including postpartum family planning. Similar demographic patterns were identified in a study published by Tocce et al., where younger medical practitioners preferred long-acting reversible contraceptives (LARCs) for postpartum family planning [32].

The knowledge regarding postpartum family planning and LARCs is prevalent among healthcare practitioners, but on the other hand, a major section of practitioners avoids answering questions related to family planning counselling and preferring LARCs immediately postpartum, highlighting various doubts related to training, institutional backing, patients’ acceptance, and cultural taboos. This aligns with the reports of Hoffman et al. and Dehlendorf et al. [33,34], who reported that the knowledge of LARCs is prevalent among practitioners and that their implementation is met with challenges due to a lack of training, institutional backing, and patient acceptance.

The study underscores that the inadequate training and unavailability of an implant, i.e., IMPLANON^®^ and PPIUCDs, are identified as key barriers in practicing immediate postpartum family planning. These findings highlight the similarities with those of Gonie et al. [35] and align with those of Ross and Keesbury, who reported that supply chain mismanagement reduces the availability of contraceptive devices at healthcare facilities when needed [36].

The fondness of the IUCD carried by healthcare practitioners and patients underscores the importance of physician and patient preferences. Similar results were reported in research published by Najan et al., who reported a high rate of willingness among women experiencing parturition to visit the IUCD postpartum [37]. On the other hand, the current study revealed that a very high percentage of patients showed a lack of interest in postpartum contraception, which can be managed through proper counselling and education. Performing continuous education programs may result in better acceptance of LARCs [38].

Furthermore, a lack of support from hospital management and strong cultural/religious myths were very clearly documented in the literature. Dismissing cultural myths [39] through continuous educational programs and improving supply chain management in hospital departments may improve both practitioners’ and patients’ perceptions of IUCDs and family planning procedures immediate to postpartum [40].

The results of the statistical analysis revealed strong associations between age, qualification, place of practice, and offering of immediate postpartum family planning. More highly qualified and experienced practitioners are likely to offer and implement LARC methods, which is similar to the findings of a study published by Rapkin et al. [41], which suggested that experienced and highly qualified professionals with better training are more proactive in implementing advanced contraceptive methods [42].

The SWOT analysis reveals that the place of practice, duration of practice, educational qualification along with the familiarities with the immediate postpartum family planning, knowledge, offerings, and counselling to the patient by the practitioner are identified as strengthens in the implementation of family planning procedures immediately postpartum. This aligns with the explanation of Ouyang M et al., which emphasizes on imparting IUCD training and its positive outcomes in developed countries globally [43].

Even practitioners with higher educational qualification and experience neglect family planning counselling, reportedly due to time constraints and unavailability of IUCDs in the practicing settings. Nevertheless, inadequate training to the relatively recent qualified practitioners regarding insertion techniques of IUCDs further slow down the practice of family planning and was identified as a weakness of the study participants. The study explains aspects of research findings similar to the study published by Mishra N et al., explaining the barriers towards the adoption of IUCDs as a proper contraceptive method [44].

Recent qualified practitioners preferred not offering family planning counselling to the patients even when having adequate time because of inadequate training and information related to the availability and technique of insertion of IUCDs, especially those posted in rural areas, highlighting the opportunities for planning better training and continuous professional development sessions for the fresh practitioners and developing, modifying, or updating the supportive policies for the family planning and IUCDs, which may improve the applicability of postpartum contraception. Similar ideas were presented by a study published by Gehani M et al., explaining the potential for improving IUCD services and applications [45].

Irrespective of the practitioner’s qualification and experience, the knowledge regarding immediate postpartum family planning counselling, PPIUCDs, and lack of proper training show a huge gap in the interest of service providers and are considered threats/drawbacks. Hence, strengthening the policies and developing the interests and awareness among practitioners may troubleshoot the issue. These obstacles have the capacity to compromise the efficient execution of postpartum contraceptive practices. Similar findings were identified in the study published by Akers AY et al., where the challenges were explained from the healthcare providers’ perspective [46].

As evident in the current study, healthcare professionals and patients require education programs reflecting the necessary skills and knowledge regarding the LARC methods and addressing the availability of contraceptives with hospital management, with the importance of improving institutional support and diminishing cultural myths through community educational programs, which may improve the acceptance of the LARC.

Future researchers should focus on longitudinal studies to determine the effects of long-term outcomes of postpartum LARC use and the success of training programs. Statistically comparative research across different regions and healthcare settings globally may provide in-depth knowledge of the factors influencing LARC implementation.

## 5. Conclusions

This research highlights various challenges hindering the application of intrauterine contraceptive devices (IUCDs) and long-acting reversible contraceptive (LARC) take-up among healthcare providers. By placing these results among the wider context of the available literature, it is evident that although awareness and approval of LARC techniques are limited by insufficient training, supply shortages and institutional and cultural hurdles need to be addressed. In practice, effective family planning relies heavily on training programs, stable supply chain, and supportive institutional regulations. Furthermore, studies must continue investigating these domains to implement the procedures of family planning effectively.

## Figures and Tables

**Table 1 healthcare-12-02208-t001:** Demographic details of the study participants.

Variables	Sub Variables	Number (%)
Age in years	20–39 years	199 (67.9)
	40–59 years	85 (29.0)
	Above 60 years	9 (3.1)
Qualification	MBBS	42 (14.3)
	FCPS part 1	36 (12.3)
	FCPS	160 (54.6)
	MCPS	39 (13.3)
	DGO	13 (4.4)
Place of practice	Teaching hospital	169 (57.7)
	District hospital	17 (5.8)
	Govt. maternity home	5 (1.7)
	THQ	38 (13.0)
	Private hospital/multidisciplinary group	34 (11.6)
	Private clinic	19 (6.5)
	DHQ	9 (3.1)
Practice location	Urban inner city	150 (51.2)
	Urban slums	26 (8.9)
	Rural	58 (19.8)
	Urban posh area	53 (18.1)
Average number of deliveries/day	0–50 deliveries per day	287 (98.0)
	51–100 deliveries per day	4 (1.4)
	101–150 deliveries per day	2 (0.7)
Average number of deliveries/month	0–500 deliveries per month	171 (58.4)
	501–1000 deliveries per month	79 (27.0)
	1001–1500 deliveries per month	30 (10.2)
	More than 1501 deliveries per month	12 (4.1)
How long have you been practicing?	1–4 years	140 (47.8)
	5–10 years	56 (19.1)
	11–15 years	28 (9.6)
	More than 15 years	63 (21.5)

**Table 2 healthcare-12-02208-t002:** Descriptive outcomes of counselling.

Variables	Sub Variables	Number (%)
Do you offer family planning counselling to patients?	Yes	276 (94.2)
	No	14 (4.8)
If yes, when do you offer counselling to patients?	Antenatal period	220 (75.1)
	Immediate postpartum	34 (11.6)
	48 h postpartum	7 (2.4)
	6 weeks postpartum	14 (4.8)
If no, why not?	Lack of time	13 (4.4)
	It is the job of the family planning department	5 (1.7)
	You do not think it is important	-
	Lack of patient interest	21 (7.2)
	Do not have education material	8 (2.7)
	Never thought about it	5 (1.7)
	Missing values—failed to answer	241 (82.3)

**Table 3 healthcare-12-02208-t003:** Descriptive outcomes of postpartum family planning.

Variables	Sub Variables	Number (%)
Are you familiar with immediate postpartum family planning?	Yes	286 (97.6)
	No	4 (1.4)
Are you offering immediate postpartum family planning to patients?	Yes	223 (76.1)
	No	58 (19.8)
If yes,		
Which one?	IUCD	154 (52.6)
	Implant	85 (29.0)
Which is the patients’ preference?	IUCD	119 (40.6)
	Implant	67 (22.9)
If no, why not?		
Lack of proper training to Implanon	Yes	98 (33.4)
	No	78 (26.6)
Lack of proper training to PPIUCD	Yes	37 (12.6)
	No	109 (37.2)
You do not have time	Yes	14 (4.8)
	No	120 (41.0)
More expulsion rate of IUCD	Yes	42 (14.3)
	No	98 (33.4)
Nonavailability of implant	Yes	173 (59.0)
	No	28 (9.6)
Nonavailability of IUCD	Yes	35 (11.9)
	No	108 (36.9)
Lack of patient interest	Yes	135 (46.1)
	No	34 (11.6)
You think IUCD is unislamic	Yes	16 (5.5)
	No	124 (42.3)
Fear of side effects	Yes	79 (27.0)
	No	81 (27.6)
You refer the patients to family planning	Yes	104 (35.5)
	No	59 (20.1)
Loss of thread	Yes	71 (24.2)
	No	79 (27.0)
Lack of support from hospital management	Yes	64 (21.8)
	No	92 (31.4)
Strong myths	Yes	95 (32.4)
	No	60 (20.5)
Lack of training in insertion	Yes	48 (16.4)
	No	106 (36.2)
**Do you think LARCs are contraindicated in;**		
Nulliparous women?	Yes	69 (23.5)
	No	201 (68.6)
Breast feeding?	Yes	9 (3.1)
	No	243 (82.9)
Immediate post placental?	Yes	12 (4.1)
	No	236 (80.5)
Postpartum?	Yes	10 (3.4)
	No	239 (81.6)
H/O ectopic pregnancy?	Yes	64 (21.8)
	No	199 (67.9)

**Table 4 healthcare-12-02208-t004:** Statistical analysis (chi-square test) of the responses of the study participants.

Variable 1	Variable 2	Chi-Square (X^2^)	df	*p*-Value	Interpretation
**Age**	Do you offer family planning? (yes/no)	7.055	2	0.029	Significant association
	If yes, do you offer immediate postpartum family planning?	12.89	4	0.045	Significant association
	If no, is it due to lack of training to PPIUCD?	6.400	2	0.041	Significant association
	If no, is it due to nonavailability of IUCD?	12.494	2	0.002	Highly significant association
	If no, is it due to referring patients to family planning centres?	12.424	2	0.002	Highly significant association
**Qualification**	Do you offer family planning? (yes/no)	39.68	4	0.000	Extremely significant association
	If yes, do you offer immediate postpartum family planning?	24.06	12	0.020	Significant association
	Familiarity with immediate postpartum family planning	12.37	4	0.015	Significant association
	Are you offering immediate postpartum family planning to patients? (yes/no)	22.18	4	0.0001	Extremely significant association
	If yes, do you offer immediate postpartum family planning to patients via IUCD?	10.87	4	0.028	Significant association
	If no, is it due to lack of proper training to Implanon?	10.11	4	0.039	Significant association
	If no, is it due to lack of proper training to PPIUCD?	41.08	4	0.000	Extremely significant association
	If no, is it due to nonavailability of IUCD?	14.16	4	0.007	Highly significant association
	If no, is it due to lack of training in insertion?	23.94	4	0.000	Extremely significant association
**Place of practice**	Do you offer family planning? (yes/no)	18.58	6	0.005	Significant association
	Are you offering immediate postpartum family planning to patients? (yes/no)	25.82	6	0.000	Extremely significant association
	If no, is it due to nonavailability of IUCD?	16.45	6	0.011	Significant association
	If no, do you refer the patients to family planning centres?	19.41	6	0.004	Significant association
	If no, is it due to lack of training in insertion?	16.35	6	0.012	Significant association
**How long have you been practicing?**	Do you offer family planning? (yes/no)	9.01	3	0.029	Significant association
	If no, is it due to lack of proper training to PPIUCD?	11.38	3	0.010	Significant association
	If no, is it due to lack of time?	7.81	3	0.050	Significant association
	If no, is it due to nonavailability of IUCD?	8.17	3	0.043	Significant association
	If no, do you refer the patients to family planning centres?	11.118	3	0.011	Significant association
**Deliveries per day**	Are you offering immediate postpartum family planning to patients like IUCD or Implant?	51.07	28	0.005	Significant association

**Table 5 healthcare-12-02208-t005:** Displaying the correlation between research variables via Pearson’s correlation coefficient (2-tailed) to facilitate thematic analysis. (SWOT)-Values (Pearson’s correlation with 2-tailed significance).

**Strengths** Knowledge and provision of immediate postpartum family planning are boosted by practitioner’s place of practice, experience, and qualification. Qualified and experienced practitioners are better at referring patients to family planning services. Qualified and experienced practitioners have better understanding of LARC contraindications and their relation to immediate postpartum family planning. Offers immediate postpartum family planning. LARCs and their contraindications. Ectopic pregnancies. Nulliparous women. Breast feeding mothers. Immediate post placental conditions.	**Weakness** Higher qualified practitioners are frequently less committed to provide family planning counselling. Higher qualified practitioners often report perceived lack of time for providing family counselling. The challenges identified include (1) Unavailability of IUCD, (2) Lack of adequate training of IUCD insertion, (3) Varying place of practice.
**Opportunities** Supplementary training programs can be planned for well-qualified and experienced practitioners. Addressing the following issues can boost the success of family planning services: (1) Training programs related to IMPLANON insertion, (2) Training programs related to IUCDs and PPIUCDs, (3) Increasing the availability of IUCDs.	**Threats** Notable gaps in training and lack of familiarity with postpartum family planning practices were identified among practitioners. Inadequate training in the insertion of PPIUCDs poses a considerable obstruction. Experienced practitioners were identified lacking interest towards family planning practice.

## Data Availability

Data from this study are available upon request from the corresponding author.

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
