# Peer review of "The Thematic Analysis of Barriers to Immediate Post-Partum Long-Acting Reversible Contraception"

_healthcare, 2024, doi:10.3390/healthcare12222208_

Round 1
Reviewer 1 Report
Comments and Suggestions for Authors
Abstract methods: consider describing the setting rather than giving the exact unit.
Abstract Methods: your title says this is a thematic analysis but there is not detail of thematic analysis in the abstract
line 113 - "dropping out or excluding"- rephrase
Lines 118 to 121 - consider breaking up into 2 sentences for clarity
Line 147 - replace ",etc" with more specifics of the way the survey was distributed
Line 171- What methods were used for the thematic analysis? What type of qualitative questions were included? How was the information coded? who did the coding? consider using the coreq checklist.
The tables need to be revised. There needs to be lines between the sections on table one. It is very difficult to read. Additionally, there are too many tables without an appropriate explanation. The table should be visually optimized. The discussion section reads more like a result section. Consider revising the results section so that it describes the results. The discussion should be a more detailed discussion of the implication of those results. For example, right now the discussion feels like a list of values rather than a true discussion.
Table 5 is unclear. I am hoping that once the method section is revised to include a Description of how the automatic analysis was performed at this table will make more sense.Author Response
please see the attachments

Reviewer 2 Report
Comments and Suggestions for Authors
Dear authors, what is the main question of your manuscript? Does this issue address to gynecologists, patients, other medical practitioners?
What brings new you manuscript in the field?
The SWOT table is hard to understand, I do not know how “ - Lack of proper training to Im planon” is an opportunity.
In the methodology I think a relevant question for the health providers would be why they did not follow training in the field? Also how would they see an improvement in the field.
The conclusions are consistent with the evidence.
The references seems appropriate.
Thank you.
Round 2
Reviewer 1 Report
Comments and Suggestions for Authors
1. ABSTRACT: "Basing up on the key statistical findings and the significance association of the responses a thematic (SWOT) based differentiation of the items was performed following the Braun and Clarke (2006) method and COREQ checklist." This sentence is unclear.
2. ABSTRACT: ". Place of practice and the experience of practitioner played a significance role in practicing, offering immediate post-partum contraception and referring the patients for family plan- 122 ning was observed as strengths." Please correct grammar.
3.line 168 - "However, data on the extent to which providers utilize these opportunities and the role of family planning counselling during antenatal care in promoting the use of postpartum modern family planning remain limited, especially in developing countries where the knowledge supported with training is evaluated, religious and cultural restrictions will be identified, impact of education, availability IUCDs and adequate policies from the perspective of practitioners will be identified imparting novelty to the study." This sentence is unclear.
4. methods - add country or city of study
5. line 186 - involve is not the correct verb. consider: "consist of"?
"the inclusion criteria involve practicing healthcare pro- 186 fessionals’ expertise in obstetrics and gynecology providing services across different geo- 187 graphical settings, thus demonstrating an intention to include a rich cohort of required 188 healthcare providers in the study". How is expertise part of your inclusion criteria? How did you define expertise?
6. line 271 - please correct grammar
7. 278- change semi colon to colon
8. line 278- "Qualitative questions included in the study are: number of deliveries in a day and in a month, practicing experience in years". These are not qualitative questions.
9. line 289- I have not seen "opted" used in this way. Confirm this word is used correctly.
10.line 292- where are items of domain 1, domain 2-item 17, and domain 2-item 18 in the paper?
11. line 328 "As many as 98.0% of the participants expressed performing 0–50 deliveries per day" Is this a typo?
12. table 3 is unreadable. Put yes/no into columns to allow for easier viewing.
13. Table 4 also needs improvement. Consider changing the variable to the statement rather than a question. I.e. How is "If yes, Which one?" a variable associated with qualification? The table should enhance understanding, but this one adds confusion.
14. Under opportunities: "• Substantial opportunities exist for enhancing family counselling". Consider removing as it does not add to table.
15. Threats: "• Acceptable experienced practitioners were identified lacking interest towards family planning practice." This sentence is unclear.
16. line 608 - fix grammar
17. line 610- repetitive language. consider rephrasing
18. line 823- replace "fresher"
19. line 856- ". In practice, training programs, disturbed supply chains, 856 and supportive institutional regulations underscore the significance of family planning 857 procedures." This sentence is unclear.
Comments on the Quality of English Languagesee above
